# Fusion of China ZY-1 02D Hyperspectral Data and Multispectral Data: Which Methods Should Be Used?

**Han Lu** [1,2,3,†]**, Danyu Qiao** [1,2,3,†]**, Yongxin Li** [4]**, Shuang Wu** [1,2,3] **and Lei Deng** [1,2,3,*]

1 College of Resource Environment and Tourism, Capital Normal University, Beijing 100048, China; 2200901010@cnu.edu.cn (H.L.); 2190902135@cnu.edu.cn (D.Q.); 2200902139@cnu.edu.cn (S.W.)
2 College of Geospatial Information Science and Technology, Capital Normal University, Beijing 100048, China
3 Key Laboratory of 3D Information Acquisition and Application, Capital Normal University, Beijing 100048, China
4 Logistics Support Department, Capital Normal University, Beijing 100048, China; yongxin@cnu.edu.cn
* Correspondence: denglei@cnu.edu.cn
† Co-first author.

**Abstract:** ZY-1 02D is China's first civil hyperspectral (HS) operational satellite, developed independently and successfully launched in 2019. It can collect HS data with a spatial resolution of 30 m, 166 spectral bands, a spectral range of 400~2500 nm, and a swath width of 60 km. Its competitive advantages over other on-orbit or planned satellites are its high spectral resolution and large swath width. Unfortunately, the relatively low spatial resolution may limit its applications. As a result, fusing ZY-1 02D HS data with high-spatial-resolution multispectral (MS) data is required to improve spatial resolution while maintaining spectral fidelity. This paper conducted a comprehensive evaluation study on the fusion of ZY-1 02D HS data with ZY-1 02D MS data (10-m spatial resolution), based on visual interpretation and quantitative metrics. Datasets from Hebei, China, were used in this experiment, and the performances of six common data fusion methods, namely Gram-Schmidt (GS), High Pass Filter (HPF), Nearest-Neighbor Diffusion (NND), Modified Intensity-Hue-Saturation (IHS), Wavelet Transform (Wavelet), and Color Normalized Sharping (Brovey), were compared. The experimental results show that: (1) HPF and GS methods are better suited for the fusion of ZY-1 02D HS Data and MS Data, (2) IHS and Brovey methods can well improve the spatial resolution of ZY-1 02D HS data but introduce spectral distortion, and (3) Wavelet and NND results have high spectral fidelity but poor spatial detail representation. The findings of this study could serve as a good reference for the practical application of ZY-1 02D HS data fusion.

**Keywords:** ZY-1 02D; hyperspectral remote sensing; multispectral remote sensing; data fusion

## 1. Introduction

The ZY-1 02D Satellite, also known as a 5-m optical satellite, is the first operational civil hyperspectral (HS) satellite, independently developed and successfully operated by China as the China–Brazil Earth Resources Satellite. It was launched on 12 September 2019, and one of the three main payloads is an advanced HS imager developed by the Shanghai Institute of Technical Physics (SITP), Chinese Academy of Sciences. The ZY-1 02D HS imager has 166 spectral bands ranging from 400 nm to 2500 nm, a spatial resolution of 30 m, and a swath width of 60 km, allowing it to provide detailed spectral information about ground features. Compared with other on-orbit or planned satellites (e.g., Environmental Mapping and Analysis Program, Precursore Iperspettrale della Missione Applicativa, and DLR Earth Sensing Imaging Spectrometer), the spectral resolution and the swath width of the ZY-1 02D are more advantageous. Unfortunately, due to the unavoidable trade-offs between spectral resolution, spatial resolution and signal-to-noise ratio [1,2], the ZY-1 02D HS data cannot directly make full use of its advantages in some specific applications [3–7]. In this case, a simple and feasible solution that improves spatial

resolution while preserving spectral fidelity is required. Nowadays, data fusion has become an important data processing method for achieving the aforementioned goal, as it can fuse HS data with MS data to combine the benefits of both [8]. Given the availability of ZY-1 02D MS data and the benefits of imaging under the same conditions as HS data, this paper focuses on fusing ZY-1 02D HS data with ZY-1 02D MS data to fully excavate and effectively use its internal information [9,10].

In recent years, with the development of high-precision quantitative remote sensing applications, a large amount of remote sensing data with both high spatial resolution and high spectral resolution has become urgently needed [11–14]. To address this issue, domestic and international researchers have conducted a lot of research in the hopes of improving spatial resolution and enriching information while preserving the spectral fidelity of the original image [15]. Data fusion technology can synthesize the effective information of various image data, eliminate the redundancy and contradictions between multiple sources of information, and produce a composite image with improved interpretability [16,17]. So far, several mature image fusion methods have been developed, including the Gram-Schmidt (GS) transform [18], Intensity-Hue-Saturation (IHS) [19], High-Pass Filter (HPF) [20], Nearest-Neighbor Diffusion (NND) [21], Wavelet Transform (Wavelet) [22], Principal Component Analysis (PCA) [23], Color Normalized Spectral Sharpening (CNSS) [24] and Color Normalized Sharping (Brovey) [25], etc. To evaluate the quality of the fused result, statistical indicators (mean, standard deviation, and mean gradient, etc.) or ground object classification accuracy, and target extraction accuracy are typically used [26–29]. Among the numerous studies, Huang et al. [30] compared the performance of HPF, IHS and super-resolution Bayesian (Pansharp) in the image fusion application of Mapping Satellite-1 (TH-1) and found that HPF is the best. Sun et al. [31] made a performance comparison of five different fusion methods to find a suitable method for GaoFen-2 (GF-2), and the results show that the fused images transformed by HCS and GS have good performance in both visual interpretation analysis and ground object classification. Du et al. [32] used four methods including Pansharp, GS and Wavelet to carry out data fusion on GaoFen-1 (GF-1), finding that the GS method could effectively improve the spatial resolution and enrich the texture information. Huang et al. [33] fused the ZiYuan-3 (ZY-3) satellite data with a variety of commonly used image fusion methods and evaluated the fusion results from both qualitative and quantitative aspects. Obviously, it can be seen that these fusion methods have good performance in high-spatial-resolution satellite data fusion. In addition, there have also been some studies considering the use of multi-source data fusion to explore the best fusion method for different data, such as Ren et al. [34], who conducted a comprehensive evaluation study on the fusion results of GF-5 HS data with three MS data (namely GF-1, GF-2 and Sentinel-2A), and the results showed that LANARAS, Adaptive Gram-Schmidt (GSA), and modulation transfer function (MTF)-generalized Laplacian pyramid (GLP) methods were more suitable for fusing GF-5 with GF-1 data, while MTF-GLP and GSA methods were recommended for fusing GF-5 with GF-2 data, and GSA and smoothing filtered-based intensity modulation (SFIM) could be used to fuse GF-5 with S2A data. Ghimire et al. [35] also created an optimal image fusion and quality evaluation strategy for various satellite image data (GF-1, GF-2, GF-4, Landsat-8 OLI, and MODIS).

The preceding studies point us in the right direction for data fusion. It can be found that different fusion methods have different fusion performance when applied to remote sensing data. This disparity can be attributed to the characteristics of remote sensing data as well as the fusion method. As a result, in practice, it is critical to select the appropriate fusion method according to the image characteristics and application purpose [36]. Given the short launch time of the ZY-1 02D satellite, there are currently few related research and results regarding image fusion of ZY-1 02D data. Because the fusion methods suitable for existing satellite data are not necessarily suitable for the ZY-1 02D, it is critical to investigate its suitable fusion methods, and this is of great significance for future application and research using ZY-1 02D data. Furthermore, when compared to multi-source remote

sensing data, ZY-1 02D provides HS and MS data under the same imaging conditions, reducing the uncertainty caused by input data. The fusion results of ZY-1 02D will have more application value. For these reasons, we will look into image fusion on ZY-1 02D HS and ZY-1 02D MS data. There are numerous fusion methods available for us to use in this research [37–42], but many of them are not user-friendly in terms of operability, computing resource requirements, and professional requirements. Taking into account the needs of users in practical applications, the focus of this research is to identify suitable methods for ZY-1 02D data among the existing well-known methods.

The objective of this paper is to find appropriate methods for fusing ZY-1 02D HS and ZY-1 02D MS data from among some well-known methods. To meet the main objective, we test six common image fusion methods (i.e., GS, HPF, IHS, Wavelet, NND, and Brovey) and use a comprehensive evaluation framework to evaluate their performance, including aspects of their visual interpretation and quantitative metrics. This study will serve as a reference for the choice of fusion methods for the ZY-1 02D data, thereby further broadening its application in a variety of fields.

## 2. Materials and Methods

### 2.1. ZY-1 02D Data

At 11:26 a.m. on 12 September 2019, the ZY-1 02D satellite was launched into the planned orbit by a Long March-4B carrier rocket from the Taiyuan Satellite Launch Center in China's Shanxi Province. It is currently in a solar synchronous orbit 778 km above the earth, with a five-year expected lifespan. The two sensors on board enable it to effectively acquire 8-band MS data with a width of 115 km and 166-band HS data with a width of 60 km. Specifically, the spatial resolution of the MS data is 10 m, the HS data is 30 m, and the spectral resolution of the HS reaches 10 nm and 20 nm in the visible-near infrared (VIS-NIR) and short-wave infrared (SWI) ranges. The main parameters of the ZY-1 02D HS and MS sensors are shown in Table 1. Figure 1 shows the spectral response function of MS sensor.

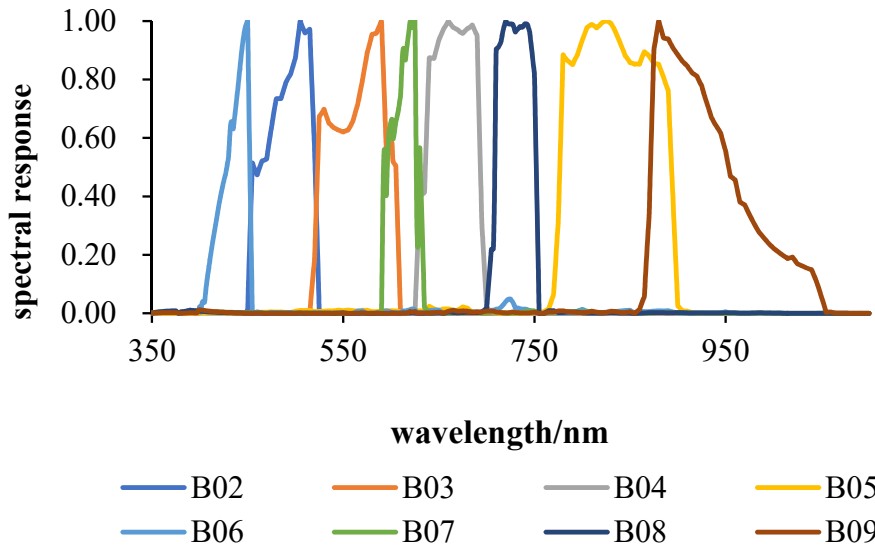

**Figure 1.** Spectral response function of multispectral sensor.

**Table 1.** Parameters of ZY-1 02D hyperspectral and multispectral sensors.

| Sensors | Bands | Spectral Range/nm | Spatial Resolution/m | Spectral Resolution/nm | Swath Width/km |
|---|---|---|---|---|---|
| MS | B02 | 452~521 | 10 | | 115 |
| | B03 | 522~607 | | | |
| | B04 | 635~694 | | | |
| | B05 | 776~895 | | | |
| | B06 | 416~452 | | | |
| | B07 | 591~633 | | | |
| | B08 | 708~752 | | | |
| | B09 | 871~1047 | | | |
| HS | VIS-NIR | 396~1039 | 30 | 10 | 60 |
| | SWI | 1056~2501 | | 20 | |

For the experiment, the ZY-1 02D HS image and MS image covering Anxin County, Baoding City, Hebei Province, China were used (Figure 2). They were collected on 7 October 2020. There are several common surface types in the area, mainly artificial buildings, farmland, vegetation, and water.

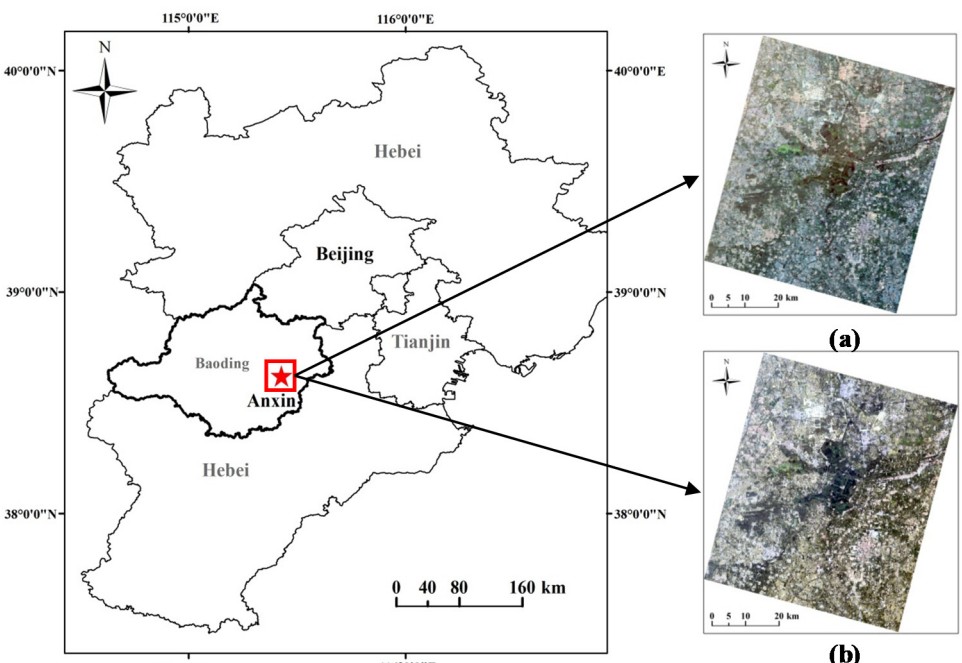

**Figure 2.** Location of study area. (**a**) hyperspectral image; (**b**) multispectral image.

*2.2. Data Preprocessing*

The preprocessing of the data was divided into two sections: (1) MS data preprocessing, and (2) HS data preprocessing. MS data preprocessing included image mosaicking, image clipping and image registration. First, the two MS images were mosaicked into one image, and then the part that overlapped with the HS image was clipped out (the size of the clipped image was 7670 × 8291 pixels). Then, the automatic registration tool in ENVI (i.e., Image Registration Workflow) was used to register it with the HS image (RMS Error: 0.31), and the processed MS image was used as input data for the fusion experiment. HS data preprocessing included radiometric calibration, atmospheric correction, and band extraction. Radiation calibration was used to covert the gray value (digital number, DN) into top of atmosphere (TOA) reflectance. FLAASH model was used for atmospheric correction. The atmospheric model was set to mid-latitude summer, and the aerosol model parameters were set to city. Since the spectral range of the HS image (396~2501 nm) was wider than that of the MS (452–1047 nm), a subset of the data spanning the range 452~1047 nm was

extracted from the HS data, and the bands were grouped to match specific MS bands (Table 2), and the processed HS image was used as the input data of the fusion experiment. All of the preceding procedures were carried out using the ENVI 5.3 software.

**Table 2.** Correspondence between the eight MS bands and seventy-five HS bands.

| Spectral Range/nm | Bands of MS (HS) |
| --- | --- |
| 452~521 | B02 (4~12) |
| 522~607 | B03 (13~18) |
| 635~694 | B04 (22~33) |
| 776~895 | B05 (40~55) |
| 416~452 | B06 (1~3) |
| 591~633 | B07 (19~21) |
| 708~752 | B08 (34~39) |
| 871~1047 | B09 (56~75) |

### 2.3. Fusion Method

In this experiment, six commonly used fusion methods were compared, namely GS, HPF, NND, IHS, Wavelet, and Brovey. The methodological flowchart is shown in Figure 3. In the fusion experiment, 8 MS bands and their corresponding 75 HS bands (as shown in Table 2) were input into different algorithms, respectively.

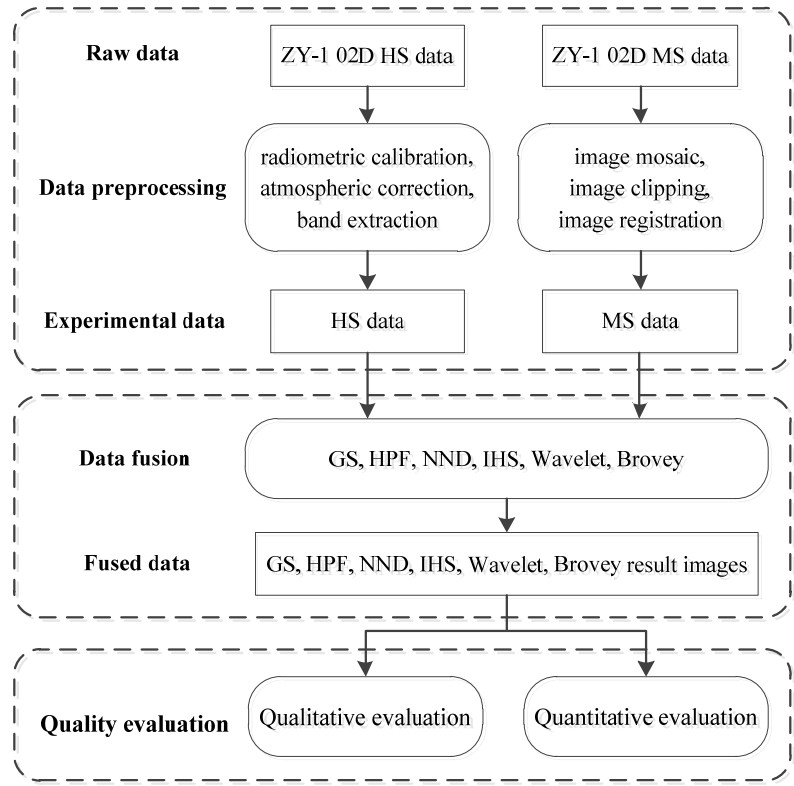

**Figure 3.** Methodological flowchart of the research. GS: Gram-Schmidt; HPF: High-Pass Filter; NND: Nearest-Neighbor Diffusion; IHS: Intensity-Hue-Saturation; Wavelet: Wavelet Transform; Brovey: Color Normalized Sharping.

### 2.3.1. Gram-Schmidt (GS)

GS transform can remove redundant information by converting an HS image to orthogonal space. The transformed components are orthogonal in the orthogonal space, and the degree of information retention varies little between them. Compared with PCA transform, this method avoids the problem of information over-concentration [43,44]. Its

advantage is that the process is simple, there are no restrictions on the number of input bands, and the spectral information of the original low-spatial-resolution image can be well preserved. During the experiment, the HS data was input as a low-resolution image, and the MS data were input as a high-resolution image (the following methods also used the same setting). The resampling method was set to nearest neighbor. This method was implemented in ENVI 5.3.

### 2.3.2. High-Pass Filter (HPF)

The HPF fusion method extracts structural detail information from the high-spatial-resolution image using a high-pass filter operator and then superimposes the detail information on the low-spatial-resolution image to achieve a combination of the two [45]. The advantages of this method include a simple algorithm, a small amount of calculation, and no limit on the number of input bands. The keral size and weighting factor were set to $5 \times 5$ and the minimum value, respectively, while the other parameters remained unchanged. This method was implemented in ERDAS IMAGINE 2014.

### 2.3.3. Nearest-Neighbor Diffusion (NND)

NND was proposed by the Rochester Institute of Technology (RIT) in the United States, which uses the Nearest-Neighbor Diffusion pan-sharpening algorithm for fusion [21]. The principle is to first perform down-sampling processing on the high-spatial-resolution image to make the spatial resolution consistent with the low-spatial-resolution image data; then, the spectral band contribution vector is calculated via linear regression, obtaining the nearest super-pixel difference factor of each pixel in the original high-spatial-resolution image; and finally, the linear mixed model is used to obtain the fused image. It is characterized by fast operation speed and high spectral fidelity. This method was implemented in ENVI 5.3.

### 2.3.4. Modified Intensity-Hue-Saturation (IHS)

IHS is a color representation system that employs intensity, hue, and saturation. When using this method for image fusion, it is primarily divided into the following three steps: (1) Resampling the low-spatial-resolution image to match the spatial resolution of the high-spatial-resolution image, then converting it from RGB space to IHS space; (2) histogram matching the high-resolution image and the I component of the low-resolution image, then replacing the I component of the low-resolution image with the new luminance component; (3) inverse transformation of the above result and its restoration to RGB space. This method has been widely used because of its high spatial detail enhancement capabilities. Compared with the traditional IHS, modified IHS overcomes the limitation of three input bands by fusing multi-band data via multiple iterations. In the B03 band, for an example, a combination of 13-14-15 and 16-17-18 is used for two iterations of fusion to produce a fused result of 13-18. The resampling method was set to nearest neighbor, and the ratio celling was set to 2.0. Because there is no ZY-1 02D in the sensor options, the band information (including center wavelength, wavelength, etc.) was customized according to the provided data file. This method was implemented in ERDAS IMAGINE 2014.

### 2.3.5. Wavelet Transform (Wavelet)

Wavelet transform is a spatial signal decomposition and reconstruction fusion technology. Its basic principle is to perform wavelet forward transformation on a low-spatial-resolution image and a high-resolution image to obtain high-frequency information from the high-resolution image and low-frequency information from the low-resolution image, respectively, and then generate a fused image using inverse wavelet transformation [46]. The spectral transform and resampling method were set to single band and nearest neighbor, respectively. This method was implemented in ERDAS IMAGINE 2014.

### 2.3.6. Color Normalized Sharping (Brovey)

The Brovey fusion method first normalizes the high-spectral-resolution data, before multiplying it by the high-spatial-resolution image to obtain the fusion result. Each band in the RGB image is multiplied by the ratio of the high-resolution data to the RGB data, and the RGB image is then resampled to the high-resolution pixel size. The resampling method was set to nearest neighbor. This method was implemented in ERDAS IMAGINE 2014.

### 2.4. Quality Evaluation Methods

We evaluate the quality of fusion results using two criteria: qualitative evaluation (i.e., visual interpretation) and quantitative evaluation (i.e., statistical metrics).

### 2.4.1. Qualitative Evaluation

Visual interpretation is the method used in qualitative evaluation, and it refers to the observer's subjective evaluation of the fusion result with respect to both the overall effect and the local effect via visual perception. Qualitative evaluation has become an indispensable part of the quality evaluation of remote sensing fusion images due to its quick and simple advantages.

### 2.4.2. Quantitative Evaluation

The use of various remote sensing image statistical metrics to evaluate the quality of the fusion results is referred to as quantitative evaluation. The advantages and disadvantages of various fusion methods can be discovered through statistics and analysis of the aforementioned metrics. In this experiment, five statistical metrics (i.e., mean, standard deviation, entropy, mean gradient, and correlation coefficient) were selected to quantitatively evaluate the fused results from the four aspects of brightness, clarity, information content, and spectral fidelity (Figure 4). The calculations of the five metrics are performed as follows.

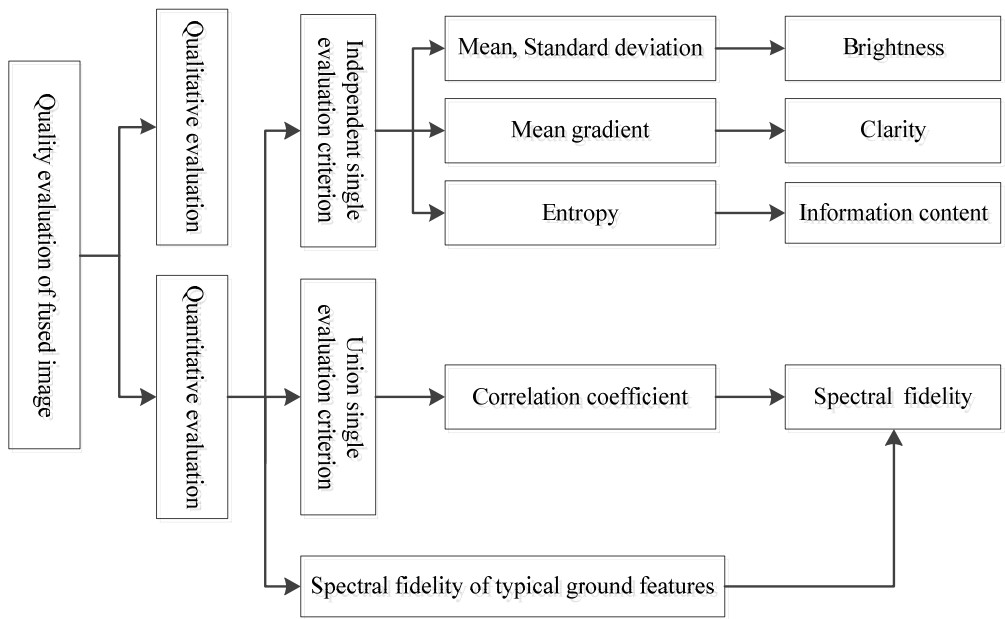

**Figure 4.** Structure of the fused image quality evaluation system.

The gray mean value is primarily used to describe the average brightness of the image. When the gray mean value of the fused image is close to that of the original multispectral image, it indicates that the fusion effect is good. It is defined as

$$\text{Mean} = \frac{1}{M \times N} \sum_{i=1}^{M} \sum_{j=1}^{N} I(i,j) \tag{1}$$

where $M$ and $N$ are the total number of rows and columns of the image, $i$ and $j$ are the pixel positions, $I(i,j)$ indicates the gray value of the pixel located in the $i$-th row and $j$-th column of the image.

The standard deviation is frequently used to describe the uniformity of image grayscale. The larger the standard deviation, the more dispersed the grayscale distribution of the image and the higher the image contrast. It is defined as

$$\text{Std} = \sqrt{\frac{1}{M \times N} \sum_{i=0}^{M-1} \sum_{j=0}^{N-1} (I(i,j) - I)^2} \tag{2}$$

where $I$ represents the gray mean value of the image.

The entropy is an important indicator for measuring the richness of image information because it reflects the average information content in the image. It is defined as

$$\text{Entropy} = - \sum P(x_i) \log(2, p(x_i)) \tag{3}$$

where $x_i$ represents the random variable, and $P(x_i)$ is the output probability function.

The mean gradient refers to the obvious difference in the gray scale near the border or both sides of the shadow line of the image, indicating that the gray scale change rate is high, and the magnitude of this change rate can be used to express the clarity of the image. It can sensitively reflect the rate at which the image expresses the contrast of small details and characterize the relative clarity and texture of the image. The larger the mean gradient, the clearer the image is. It is defined as

$$G = \frac{1}{(M-1)(N-1)} \sum_{i=1}^{M} \sum_{j=1}^{N} \sqrt{\frac{\left(\left(\frac{\partial Z(x_i, y_j)}{\partial x_i}\right)^2 + \left(\frac{\partial Z(x_i, y_j)}{\partial y_j}\right)^2\right)}{2}} \tag{4}$$

where $\frac{\partial Z(x_i, y_j)}{\partial x_i}$ represents the gradient in the horizontal direction, and $\frac{\partial Z(x_i, y_j)}{\partial y_j}$ represents the gradient in the vertical direction.

The correlation coefficient indicates how similar the images were before and after fusion. A high correlation coefficient indicates that the fused image is close to the original image and has good spectral fidelity. It is defined as

$$C = \frac{\sum_{i=1}^{M} \sum_{j=1}^{N} \left[R(i,j) - \bar{F}_R\right] \left[F(i,j) - \bar{F}\right]}{\sqrt{\sum_{i=1}^{M} \sum_{j=1}^{N} \left[R(i,j) - \bar{F}_R\right]^2 \sum_{i=1}^{M} \sum_{j=1}^{N} \left[F(i,j) - \bar{F}\right]^2}} \tag{5}$$

In addition to the statistical metrics listed above, the spectral curve of typical ground objects can also be used to evaluate the quality of the fusion results [47]. In this experiment, the spectral curves of typical ground objects (vegetation, water, and artificial building) are extracted, and a comparison is drawn between the original HS image and the fused image, respectively, to quantify the benefits and drawbacks of the fusion methods.

## 3. Results

### 3.1. Qualitative Evaluation

Figure 5 shows false-color images of the original HS image, the six fused results (R: 954 nm; G: 765 nm; B: 482 nm), and the original MS image (R: B09; G: B08; B: B02). There are some color differences between different fused images and the original HS image from the perspective of the entire image, but they all retain the main spectral characteristics of the original HS image. The colors of the HPF and Wavelet images are the closest to the original HS image, and the tone of the two is lighter than that of the original HS image, with almost no difference between the HPF and the Wavelet images. When compared

to the original HS image, the color of the NND and GS images is more orange, and the contrast between adjacent objects in the two is not as clear, indicating that the NND and GS fusion methods perform worse than the above two methods. The IHS image and the original HS image have a distinct spectral difference, which is reflected in the darker tone of the IHS image as well as the spectral distortion phenomenon. Nevertheless, IHS images possess clarity. The color deviation between the Brovey image and the original HS image is the greatest, indicating that the Brovey image has significant spectral distortion.

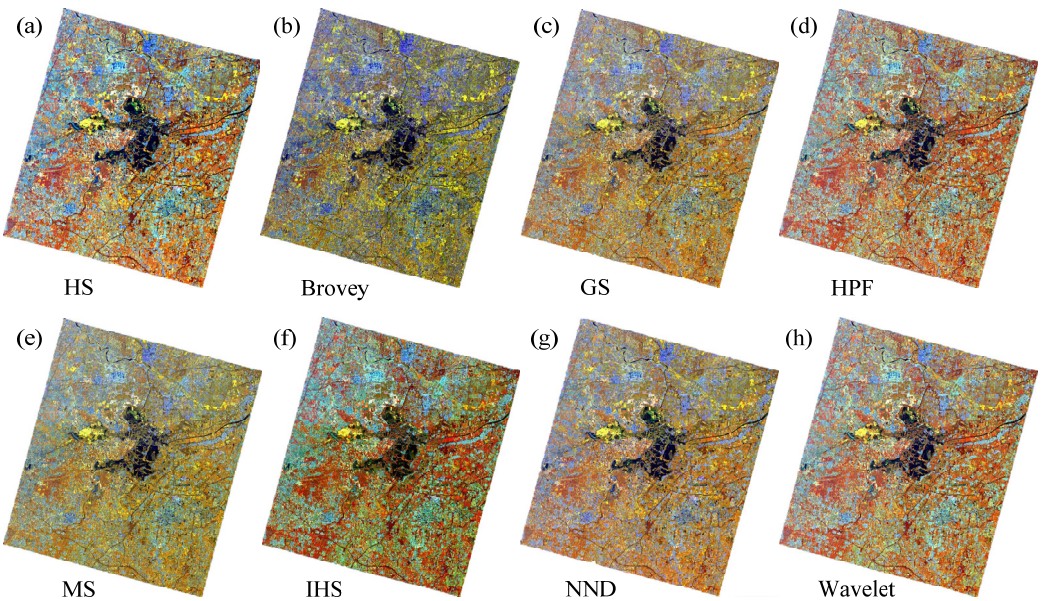

**Figure 5.** Experimental results of ZY-1 02D hyperspectral and multispectral data fusion. (**a**) ZY-1 02D hyperspectral image; (**b**) Brovey image; (**c**) GS image; (**d**) HPF image; (**e**) ZY-1 02D multispectral image; (**f**) IHS image; (**g**) NND image; (**h**) Wavelet image.

To further evaluate the visual interpretation effect of the fused images, especially the enhancement effect of the spatial details, Figures 6 and 7 show the detail of several typical ground objects, namely artificial buildings, farmland, vegetation and water.

When the edge and texture differences between the original HS image and fused image are compared, it can be found that, with the exception of the Wavelet image, the clarity of the other five fused images is higher than that of the original HS image, indicating that the above five fusion methods are able to improve the spatial resolution of the original HS image, thereby improving the accuracy and reliability of the visual interpretation. The IHS image has the best clarity and good spatial sharpening effect. The boundaries between building, road, and farmland are the clearest, and the outlines of vegetation and artificial fences in the water are the most visible, indicating that the IHS fusion method improves the spatial details of the original image the most. Unfortunately, the IHS image contains some spectral distortions, resulting in significant color differences between the IHS image and the original HS image. HPF and GS fusion methods are second only to IHS in terms of spatial detail enhancement. Specifically, the contours of aquatic vegetation are more visible in the HPF fused image, and the details of some buildings and farmland features are blurred. The detailed spatial information of the Brovey fused image is slightly lost, which is reflected in the fact that the artificial fence edge in the water is difficult to identify, and its spectral distortion is more pronounced. The spatial resolutions of the NND and Wavelet fused images are low, and the visual interpretation effect is not optimal. Between them, the Wavelet image has the closest color match to the original HS image, but the effect of its spatial detail representation is poor, and there are some obvious distortions and unclear texture features in the buildings.

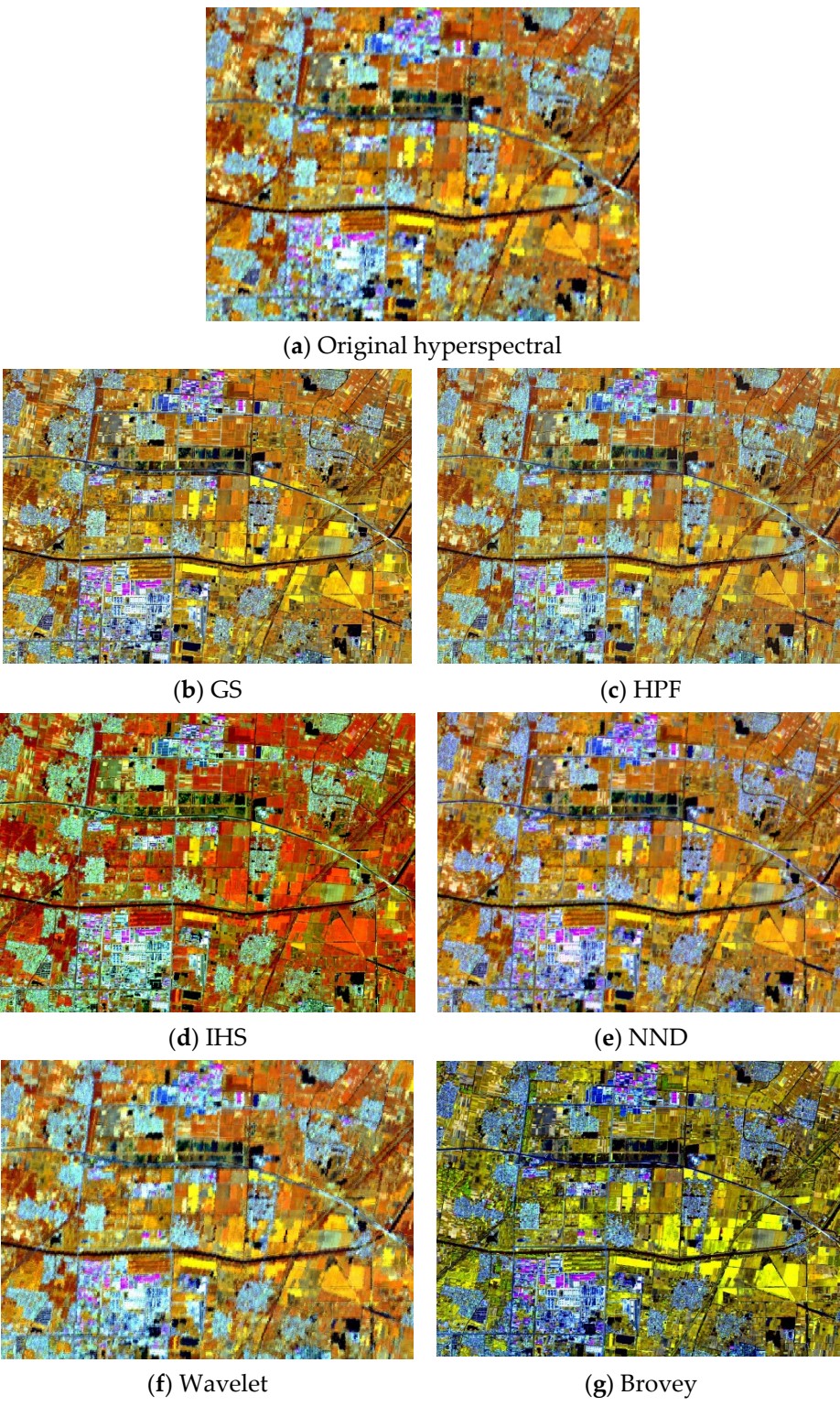

**Figure 6.** The details of farmland and artificial buildings. (**a**) ZY-1 02D original hyperspectral image; (**b**) GS image; (**c**) HPF image; (**d**) IHS image; (**e**) NND image; (**f**) Wavelet image; (**g**) Brovey image.

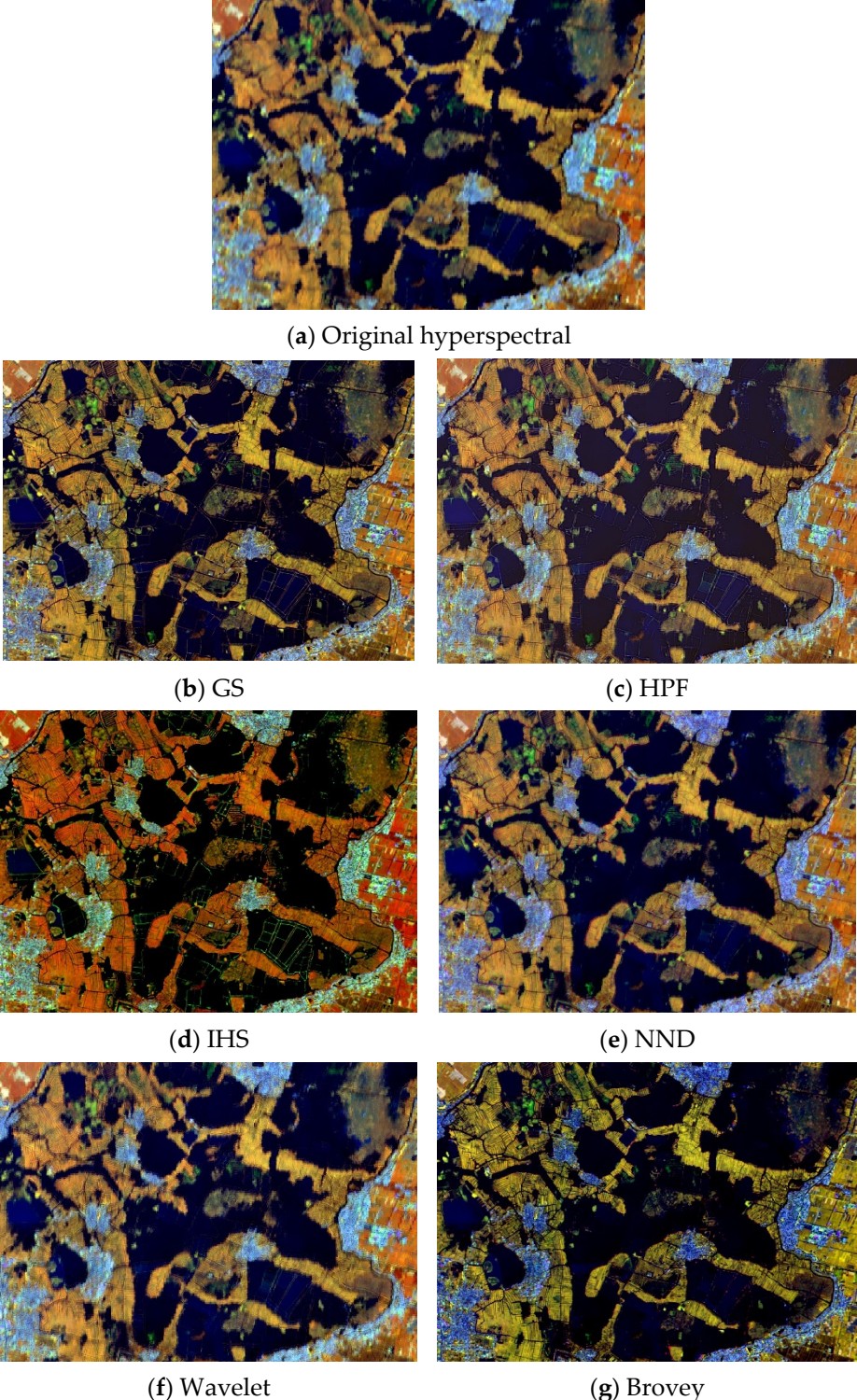

**Figure 7.** The details of vegetation and water.(**a**) ZY-1 02D original hyperspectral image; (**b**) GS image; (**c**) HPF image; (**d**) IHS image; (**e**) NND image; (**f**) Wavelet image; (**g**) Brovey image.

*3.2. Quantitative Evaluation*

3.2.1. Statistical Metrics

Considering a large number of HS bands, eight HS bands are quantitatively accounted for this experiment, corresponding to the center wavelength positions of the eight MS

bands. Table 3 shows the mean, standard deviation, entropy, mean gradient, and correlation coefficient of the original HS image and six fused images.

**Table 3.** Quantitative evaluation statistics metrics.

| Number of Bands (Wavelength) | Fusion Method | Mean | Standard Deviation | Entropy | Mean Gradient | Correlation Coefficient |
|---|---|---|---|---|---|---|
| B02 (482 nm) | Original HS | 100.3648 | 12.7672 | 6.0404 | 26.8851 | 1.0000 |
| | GS | 96.9338 | 19.6723 | 6.1241 | 47.8024 | 0.9027 |
| | HPF | 103.5575 | 24.3926 | 6.0896 | 54.4626 | 0.9448 |
| | IHS | 73.1858 | 24.1157 | 5.2708 | 49.4911 | 0.8613 |
| | NND | 103.8201 | 14.2127 | 6.0183 | 29.0472 | 0.9237 |
| | Wavelet | 102.7105 | 18.2765 | 6.1112 | 38.4877 | 0.9711 |
| | Brovey | 29.6865 | 6.3760 | 4.9725 | 46.1580 | 0.7841 |
| B03 (568 nm) | Original HS | 109.7536 | 14.0712 | 6.0792 | 30.4275 | 1.0000 |
| | GS | 108.7666 | 21.017 | 6.0606 | 54.8879 | 0.8979 |
| | HPF | 113.8481 | 25.3081 | 6.1309 | 58.9571 | 0.9479 |
| | IHS | 104.0233 | 22.5063 | 6.0326 | 52.4368 | 0.8986 |
| | NND | 111.2329 | 15.7797 | 6.0458 | 32.9132 | 0.9501 |
| | Wavelet | 112.7678 | 19.1729 | 6.1322 | 41.9692 | 0.9682 |
| | Brovey | 33.6375 | 6.7881 | 5.037 | 51.8042 | 0.7923 |
| B04 (662 nm) | Original HS | 105.6488 | 15.0996 | 6.1017 | 32.1022 | 1.0000 |
| | GS | 103.7238 | 22.5502 | 6.1283 | 55.0825 | 0.8966 |
| | HPF | 108.9427 | 26.6834 | 6.1284 | 60.8555 | 0.9426 |
| | IHS | 104.1239 | 23.5609 | 6.0404 | 54.7468 | 0.8995 |
| | NND | 108.6382 | 16.4924 | 6.0502 | 33.9025 | 0.9411 |
| | Wavelet | 111.4921 | 20.2655 | 6.1502 | 44.3468 | 0.9664 |
| | Brovey | 32.2769 | 7.2767 | 5.0268 | 53.6661 | 0.7904 |
| B05 (834 nm) | Original HS | 148.1834 | 10.9544 | 5.7649 | 24.2174 | 1.0000 |
| | GS | 147.7579 | 17.8605 | 5.9226 | 56.4583 | 0.9281 |
| | HPF | 148.4721 | 19.7851 | 6.0258 | 49.2987 | 0.9095 |
| | IHS | 129.5685 | 23.5102 | 5.8911 | 56.8161 | 0.9252 |
| | NND | 138.3355 | 14.0712 | 5.8797 | 30.2696 | 0.9403 |
| | Wavelet | 156.0215 | 16.0169 | 5.8548 | 34.2237 | 0.9813 |
| | Brovey | 48.5566 | 5.9984 | 4.9162 | 47.4623 | 0.8193 |
| B06 (430 nm) | Original HS | 98.2831 | 11.9164 | 6.0626 | 25.3595 | 1.0000 |
| | GS | 104.1284 | 15.3213 | 6.1055 | 41.8993 | 0.8887 |
| | HPF | 101.6062 | 24.5561 | 6.0715 | 55.1704 | 0.9341 |
| | IHS | 79.5842 | 22.2562 | 5.6839 | 44.5485 | 0.8834 |
| | NND | 115.3039 | 19.2873 | 6.0474 | 45.6413 | 0.8802 |
| | Wavelet | 99.2973 | 17.6968 | 6.0998 | 37.4642 | 0.9696 |
| | Brovey | 33.4879 | 5.6296 | 5.0001 | 43.9359 | 0.7811 |
| B07 (611 nm) | Original HS | 107.9341 | 14.5122 | 6.0967 | 31.1587 | 1.0000 |
| | GS | 105.9264 | 22.2558 | 6.1283 | 54.5936 | 0.8972 |
| | HPF | 110.5961 | 26.2106 | 6.1309 | 60.1416 | 0.9442 |
| | IHS | 104.6792 | 22.3629 | 6.0433 | 55.7009 | 0.8985 |
| | NND | 109.8163 | 15.9687 | 6.0527 | 33.1166 | 0.9502 |
| | Wavelet | 112.5107 | 20.2215 | 6.1508 | 44.2971 | 0.9688 |
| | Brovey | 32.2255 | 7.2689 | 5.0279 | 53.6364 | 0.8894 |
| B08 (765 nm) | Original HS | 147.6844 | 11.2318 | 5.7661 | 24.9981 | 1.0000 |
| | GS | 147.2125 | 17.7753 | 5.9252 | 57.0881 | 0.9289 |
| | HPF | 139.327 | 21.3667 | 6.0161 | 53.9328 | 0.8806 |
| | IHS | 116.0493 | 22.1731 | 6.0229 | 55.1546 | 0.6981 |
| | NND | 127.7334 | 21.6202 | 5.9381 | 56.6921 | 0.8184 |
| | Wavelet | 134.6255 | 20.4686 | 6.0776 | 45.9071 | 0.8207 |
| | Brovey | 48.9651 | 6.0844 | 4.9028 | 56.1831 | 0.8106 |
| B09 (954 nm) | Original HS | 154.8327 | 9.9505 | 5.7401 | 22.1483 | 1.0000 |
| | GS | 153.1369 | 18.2631 | 5.8793 | 51.5527 | 0.9529 |
| | HPF | 159.3981 | 18.0164 | 6.0258 | 47.0832 | 0.9775 |
| | IHS | 130.1481 | 22.2889 | 5.8911 | 55.1831 | 0.9226 |
| | NND | 146.2381 | 12.7801 | 5.8583 | 27.7978 | 0.9461 |
| | Wavelet | 159.3499 | 15.8887 | 5.8548 | 36.2066 | 0.9832 |
| | Brovey | 48.0108 | 6.1852 | 4.8995 | 46.1493 | 0.8071 |

By comparing the statistical metric values of the six fused images in Table 3 with the original HS image, the following five results can be obtained.

(1)   The HPF and GS fused images (gray mean value of the HPF image: 101~160; GS: 96~154) had high brightness similarity with the original HS image (100~155); the NND image (103~147) and Wavelet image (99~166) were second to them; the IHS image (73~131) had some spectral distortion when compared to the original HS image; the mean value of the Brovey image (29~49) was much lower than that of the original HS image, the mean value between the two had the largest deviation, and the spectral distortion of the Brovey image was the most significant.

(2)   The standard deviations of HPF, IHS and GS fused images were relatively large. Among them, the HPF image had the largest standard deviation in the visible light (standard deviation of the HPF fused image: 23~27), and the IHS image had the biggest standard deviations in the near-infrared and red edge bands (IHS image: 22~24), and the standard deviations of the three fused images were higher than the original HS image, greatly improving the information richness of the original HS image; the standard deviations of the NND and Wavelet images were slightly lower than those of the above three fused images. The standard deviation of the Wavelet image (16~21) was slightly higher than that of the NND (12~22); the standard deviation of the Brovey image (6~8) was much lower than that of the original image, indicating that the gray level of the fused image after Brovey transformation was not sufficiently dispersed and the tone tended to be single.

(3)   Except for the Brovey image, the entropy of the other five fused images was improved when compared to the original HS image, and there was no significant difference in the entropy of different fusion images (5–7). Among them, the entropy of the Wavelet image was the highest, followed by the HPF and GS images. The information content of the three was nearly identical, with the exception of the IHS, which was slightly lower; the information content of Brovey fusion images was lower than that of the original HS images.

(4)   The mean gradient of the fused image was higher than that of the original HS image, indicating that the six fusion methods were able to improve the original image's ability to represent spatial details. The mean gradients of HPF and IHS fused images were larger, and their spatial detail information enhancement effects were the best, as shown in Table 3. Between them, the mean gradient of HPF image was more visible in the visible light band (mean gradient of HPF fusion image: 47~61). The IHS fusion image outperformed the other two fusion methods in the red edge and near-infrared bands (55~57); the mean gradient of GS and Brovey fused images were slightly lower than the above two fused images. Between them, the mean gradient of Brovey image (43~57) was slightly higher than GS image (41~58); the mean gradient of NND and Wavelet fused was the lowest, indicating that the spatial resolution of these two fused images was low.

(5)   Similar to the visual interpretation result, the correlation coefficients of Wavelet, HPF and NND were relatively large, indicating that they have excellent spectral fidelity performance. Except for the red band, the correlation coefficients of Wavelet and HPF fused images were all greater than 0.9; the correlation coefficient between the IHS fusion image and the original image was second only to HPF and Wavelet, but it performed poorly in the blue and red bands; and the Brovey fused image had the lowest correlation coefficient, indicating that there was a large spectral difference between it and the original HS image.

To summarize, the HPF image not only preserved the spectral characteristics of the original HS image to a large extent, but also improved the spatial resolution of the HS data, making it the best choice for ZY-1 02D HS data enhancement; although Wavelet and NND fused images had a large amount of information and a high spectral fidelity, the spatial detail representation effect of the two was not satisfactory; the IHS fused image was clear and rich in texture information, but there was obvious spectral distortion; the performance of the GS fusion image was at a medium level; the Brovey transform had

a mediocre performance in spectral fidelity and spatial detail representation, and had a serious spectral distortion problem, making it far inferior to the above five methods.

### 3.2.2. The Spectral Curves of Typical Ground Objects

To more intuitively evaluate the spectral fidelity of the different fused images, this experiment takes typical objects (i.e., vegetation, artificial building, and water) as examples to compare the differences between the six fused results and the original HS image. To ensure that the results are representative, the spectral curves of the three objects are averaged over three homogeneous areas with 5 × 5 pixels. As shown in Figure 8, the left column is the three spectral curves of six fused images and HS image. In addition, the spectral differences between the six fused images and the original HS image are also shown (the right column of Figure 8), which is convenient for identifying the differences in the spectral fidelity of the different fusion methods.

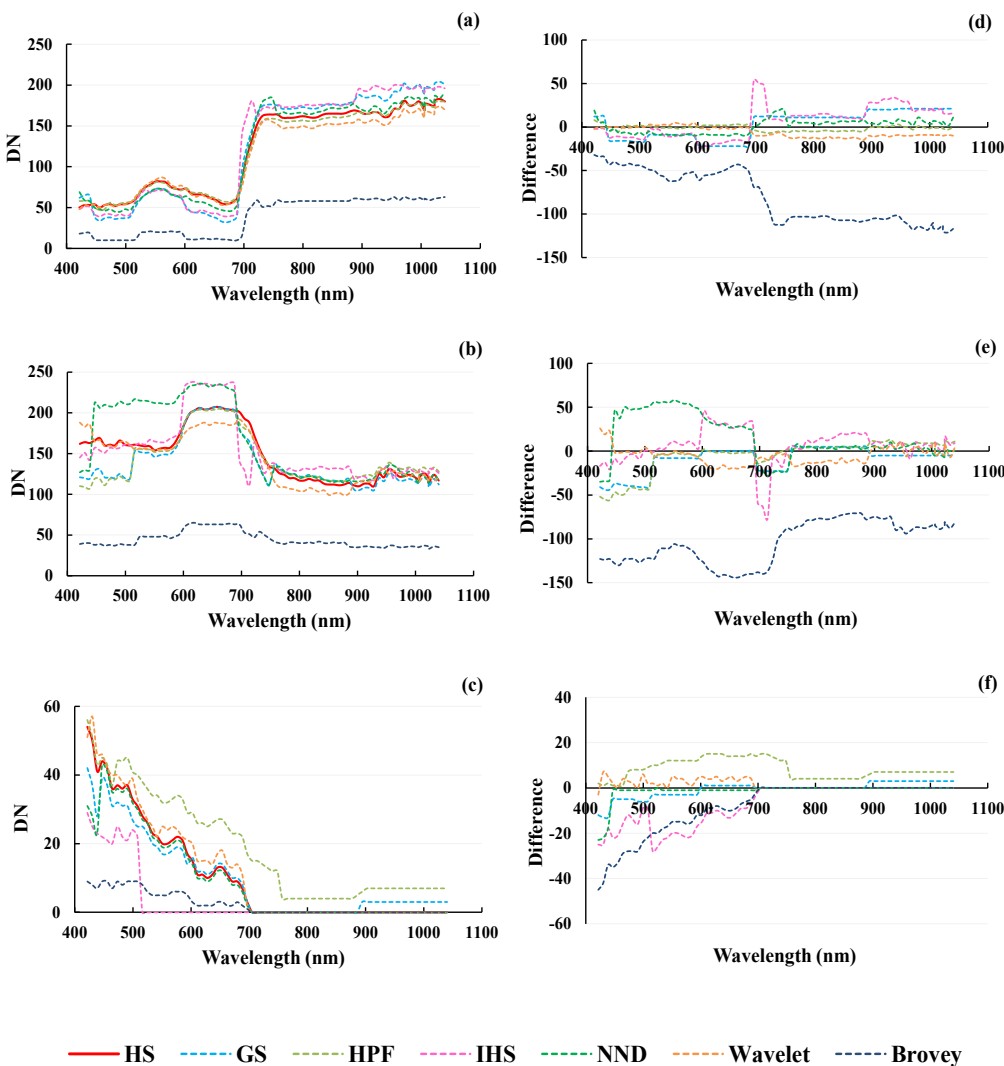

**Figure 8.** Spectral curves of typical objects. Top: vegetation spectral curves of six fused images and HS image (**a**), difference between fused images and HS image in vegetation spectral curve (**d**); Middle: artificial building spectral curves of six fused images and HS image (**b**), difference between fused images and HS image in building spectral curve (**e**); Bottom: water spectral curves of six fused images and HS image (**c**), difference between fused images and HS image in water spectral curve (**f**).

By comparing the spectral curves of different fused images and the original HS image, we can obtain similar results to those presented above. Overall, the HPF, Wavelet, NND

and GS fused images possessed high spectral fidelity; while the IHS and Brovey fused images exhibited significant spectral distortion. The spectral curves of the fused images have roughly the same trend as the original HS image and fluctuate around the original HS spectral curve with slight variations. In the spectral curves of vegetation, the difference between the Wavelet and HPF fused image and the original HS image was the smallest, indicating that the two fusion methods had the best spectral fidelity. Between them, the Wavelet fused image was more prominent in the visible light band, while the HPF fused image was more prominent in the visible light and near-infrared (NIR) bands; the spectral fidelity of the NND and GS fused images were second, with a lager difference between their spectral curves. In the spectral curves of the building, the Wavelet fused image had a higher degree of overlap and a smaller difference with the original HS image; the spectral curves of HPF and GS fused images basically fitted the original HS image after about 510 nm; and the performance of NND was quite different in the NIR band (approximately 765 nm). The curve in the second half of the spectral curve was very similar to that of the original HS image, whereas the curve in the first half was quite different from the gray value of the original HS image. This result is consistent with the color deviation of the building in the visual interpretation effect. In the spectral curves of water, the NND and the original HS image were highly overlapped; the difference between Wavelet and GS was slightly larger; the change trend of the HPF fused image was the same as the original image, but compared to the above fused methods, there was a larger difference between HPF and the original HS, which is slightly different from the high correlation coefficient obtained for the HPF method in the metric statistical results. This may have been caused by the small amount of water in the entire image. Similar to the visual interpretation results, the IHS and Brovey fused images were significantly different from the original HS image with respect to the spectral curves of the three ground objects. There were obvious spectral distortions in the two fused images, particularly the Brovey image, which had the lowest spectral fidelity.

In conclusion, the HPF, Wavelet, NND, and GS fusion methods performed well, and different fusion methods performed differently when confronted with different ground objects. HPF and Wavelet had the best spectral fidelity in the vegetation and building areas, respectively, while NND had the best spectral fidelity in the water area. It can be found that the HPF, Wavelet, NND and GS fusion methods have good performance, and different fusion methods have different performance when facing different ground objects. Specifically, HPF and Wavelet have the best spectral fidelity in vegetation and building areas, and NND has the best spectral fidelity in the water area.

### 4. Discussion

#### 4.1. Performance of Fusion Methods

In this paper, six well-known fusion methods were successfully used to enhance the spatial resolution and the information of ZY-1 02D data. Compared to the original HS data, the six fused images were clearer and easier to visually interpret. Furthermore, there were also some noticeable differences among the six fused results. HPF, in particular, demonstrated excellent performance with respect to both spectral fidelity and spatial resolution enhancement. The reason for this could be that the HPF injects the spatial details of the MS data into the HS data via a high-pass filter and has a low-pass filter to maintain the spectral separation of HS data. This result was also discovered in the fusion experiment of Mapping Satellite-1 by Huang et al. [30]. The difference is that the research object of this study is HS data with high spectral resolution, which are better suited to narrow-band spectroscopy research. Aside from HPF, Wavelet and NND exhibited high robustness in terms of spectral fidelity. Between them, the spatial resolution of the Wavelet fused image was slightly lower, and the boundary between different features was not significant, but it was always clearer than the original HS data. As concluded by Sun et al. [31] and Du et al. [32], GS performs well when applied to remote sensing data. On the contrary, the spectral fidelity performances of IHS and Brovey were slightly inferior, and there is still

room for improvement in spatial resolution [48,49]. The spectral distortion of IHS is mainly caused by the forced direct replacement of the I component [50], whereas the spectral distortion of Brovey was affected by the simple multiplication of HS and MS [25]. It is worth noting that these two methods are simple in terms of theory, simple to implement, and quick to calculate, and so can be used for some low-demand applications.

### 4.2. Selection of Quantitative Metrics

The quality evaluation of the fused results in remote sensing data fusion research is too complicated to establish a unified standard. Until now, it has been performed using a combination of visual interpretation and quantitative metrics [51]. As a result, we used the same method to evaluate the quality of six fused images. The five quantitative metrics selected for this paper adhered to three principles: (1) they were able to evaluate the quality of fused images with respect to different aspects (typically spatial resolution enhancement capability and spectral fidelity); (2) they were able to distinguish the performance differences of the six fusion methods; and (3) they were simple and easy to calculate. The comparison results also demonstrate the applicability of the five metrics. In addition to the common metrics listed above, many researchers have attempted to supplement the quality evaluation system by improving or proposing new metrics [52–54]. There is still a long way to go before we establish a unified evaluation standard. In practical applications, appropriate quantitative metrics should be selected based on geographic conditions, application requirements, and data sources.

### 4.3. Limitations

The study was carried out on a single date in a single study area with four typical surface types; it would be preferable if different study areas with more surface types in different periods were used [55]. Because the spectral range of the HS image is wider than that of the MS, this study only performed fusion processing on the HS data in the 452~1047 nm range. In subsequent research, we will fully exploit the advantages of the wide spectral range to investigate suitable fusion methods for shortwave of infrared band, as well as extending the practical application of ZY-1 02D data. In reality, the primary purpose of data fusion is to prepare for remote sensing image applications such as image classification [56], change detection [57], and so on. As a result, more research into the application of fused images is required in the future to obtain better results and conclusions.

### 5. Conclusions

Six fusion methods, GS, HPF, IHS, NND, Wavelet, and Brovey, were used in this paper to realize the fusion of ZY-1 02D HS data and MS data. The six fusion results were compared and analyzed using a combination of qualitative and quantitative evaluation methods, and the following conclusions were drawn: (1) Considering the three aspects of visual effect, spectral fidelity, and spatial detail expression, the HPF method was the most suitable for the fusion of ZY-1 02D HS data and MS data. In comparison to the original HS image, the HPF fused image maintained its spectral characteristics while improving its spatial resolution, enriching its information, and providing the best fusion performance. (2) Different fusion methods perform differently for different datasets. In practice, appropriate data fusion methods should be selected according to the data type and specific needs. Six commonly used fusion methods were used in this study for fusion processing of ZY-1 02D satellite HS and MS images, providing a significant reference for future ZY-1 02D data processing and application-related research.

**Author Contributions:** H.L., D.Q. and L.D. designed and developed the research idea. H.L., D.Q. and S.W. conducted the field data collection. H.L., D.Q., Y.L. and S.W. processed all remaining data. H.L. and D.Q. performed the data analysis and wrote the manuscript. H.L., D.Q., Y.L., S.W. and L.D. contributed to result and data interpretation, discussion, and revision of the manuscript. All authors have read and agreed to the published version of the manuscript.

**Funding:** This research was funded by the landscape plant maintenance monitoring and intelligent diagnosis model development of Capital Normal University (no. 21220030003).

**Acknowledgments:** The authors are very thankful for Zou Hanyue, Fan Tianxing and Chen Yong for their valuable support.

**Conflicts of Interest:** The authors declare no conflict of interest. The funders had no role in the design of the study; in the collection, analyses, or interpretation of data; in the writing of the manuscript, or in the decision to publish the results.

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
