# Peer review of "Fusion of China ZY-1 02D Hyperspectral Data and Multispectral Data: Which Methods Should Be Used?"

_remotesensing, doi:10.3390/rs13122354_

Round 1
Reviewer 1 Report
Fusion appears to be a tremendous amount of work for questionable results. It is very helpful to have immediate, visual geospatial referencing between the hyperspectral and multispectral data sets, but there is no real possibility of retrieving high spectral detail at the higher spatial resolution. More problematic is the possibility of using the combination as if the high spectra-spatial resolution data were a true representation of reality.
The description of the process, while well-referenced, is not described in a way that I find helpful. My main question is how the hyperspectral bands are selected for the operation. I assume that the fit is made using only the bands most closely corresponding to the center wavelengths of the MS sensor, since that is cited as the mode for evaluation. (The variant being the HIS method which is limited to 3-bands at a time.) Presumably, those 8 bands are assigned values at the higher spatial resolution, and then serve as tie points for the detailed spectrum. Whatever the case, the procedure should be summarized clearly and succinctly. Assume that there will be individuals like myself who are well versed in spectral remote sensing who are not familiar with this type of fusion.
The spectra shown in Figure 8 are completely unreliable. Even if they match at the tie points, the intervening spectral shape is useless to anyone seriously interested in spectral properties. This is not a surprise and is useful information, but should be a part of the conclusions.
Specific Comments:
Line 63: be è been
Line 69: Among the lots of research, [30] è Huang et al. [30]
Line 73: show è showing
Line 83: [34] conducted è [34] conducted
Line 93: in different è when applied to
Line 104: fusion (redundant)
Line 115: works in the è occupies a
I stopped editing. Have a native English speaker with knowledge of the technical jargon review this for proper usage. The text is intelligible, but awkward, and can be misleading.
Author Response
Dear Reviewer:
Thank you very much for your careful review and constructive comments with regard to our manuscript “Fusion of China ZY-1 02D Hyperspectral data and Multispectral Data: Which Methods Should Be Used?” (ID: remotesensing-1238235).
Those comments are all valuable and very helpful in revising and improving our paper, as well as guiding significance to our research. We carefully considered the comments and made correction that we hope will be accepted.
Our main corrections in the manuscript and response to reviewers’ comments are stored in two word files named "remotesing-1238235-revision" (main corrections are marked in red) and "response-1".
We appreciate your warm work earnestly, and hope the correction will meet with approval. Once again, thank you very much for your comments and suggestions.

Reviewer 2 Report
Authors try to determine which method should be used for best fusion of China ZY-1 02D Hyperspectral data and Multispectral Data- Paper is well written and experiments are reproducible. However, from my own point of view, paper presents some serious drawbacks that lead to reject it:
1) Although the topic is of interest, the data set is limited to one image. Authors argue that this image covers a variety of surface types in the area. However, the areas that they zoom in the paper do not seems to support this asseveration.
2) Methodology applies several well-known algorithms. Most of them have several improved versions that authors do not consider. See for example:
Mario Lillo‐Saavedra & Consuelo Gonzalo (2006) Spectral or spatial quality for fused satellite imagery? A trade‐off solution using the wavelet à trous algorithm, International Journal of Remote Sensing, 27:7, 1453-1464, DOI: 10.1080/01431160500462188
3) The quantitative evaluation is conducted also by common assessments. Some modern quality metrics should be applied. See for example:
Rodríguez-Esparragón, J. Marcello-Ruiz, A. Medina-Machín, F. Eugenio-González, C. Gonzalo-Martín and A. García-Pedrero, "Evaluation of the performance of spatial assessments of pansharpened images," 2014 IEEE Geoscience and Remote Sensing Symposium, 2014, pp. 1619-1622, doi: 10.1109/IGARSS.2014.6946757.
Rodríguez-Esparragón, D., Marcello, J., Eugenio, F., García-Pedrero, A., & Gonzalo-Martín, C. (2017). Object-based quality evaluation procedure for fused remote sensing imagery. Neurocomputing, 255, 40-51.
4) A the end, the most relevant is that results can be derived from a review of the algorithms performance literature.
Author Response
Dear Reviewer:
Thank you very much for your careful review and constructive comments with regard to our manuscript “Fusion of China ZY-1 02D Hyperspectral data and Multispectral Data: Which Methods Should Be Used?” (ID: remotesensing-1238235).
Those comments are all valuable and very helpful in revising and improving our paper, as well as guiding significance to our research. We carefully considered the comments and made correction that we hope will be accepted.
Our main corrections in the manuscript and responses to reviewers’ comments are stored in two word files named "remotesing-1238235-revision" (main corrections are marked in red) and "reviewer-2".
We appreciate your warm work earnestly, and hope the correction will meet with approval. Once again, thank you very much for your comments and suggestions.

Reviewer 3 Report
In this work, the authors have conducted a comprehensive evaluation study on the fusion of ZY-1 02D HS data with ZY-1 02D MS data (10m spatial resolution), based on visual interpretation and quantitative metrics. My comments are as follows:
- Novelty is nil as some of the existing and quite old approaches are implemented on a new dataset.
- The authors should consider recent related works. One of the suggested paper is: Multi-Scale RoIs Selection for Classifying Multi-spectral Images
- The authors have used quite old fusion metrics for comparison purpose. Thus, the authors are requested to use the following metric: Edge Information based Image Fusion Metrics using Fractional Order Differentiation and Sigmoidal Functions
Author Response
Dear Reviewer:
Thank you very much for your careful review and constructive comments with regard to our manuscript “Fusion of China ZY-1 02D Hyperspectral data and Multispectral Data: Which Methods Should Be Used?” (ID: remotesensing-1238235).
Those comments are all valuable and very helpful in revising and improving our paper, as well as guiding significance to our research. We carefully considered the comments and made correction that we hope will be accepted.
Our main corrections in the manuscript and responses to reviewers’ comments are stored in two word files named "remotesing-1238235-revision" (main corrections are marked in red) and "reviewer-3".
We appreciate your warm work earnestly, and hope the correction will meet with approval. Once again, thank you very much for your comments and suggestions.

Round 2
Reviewer 1 Report
General Comments:
This paper explores the application of several different data fusion methods applied to hyperspectral data. It is an interesting idea, and worth publishing if only to document the exploration. For me, it confirms that the spectral details of the fused data cannot be trusted quantitatively. The primary reason for using hyperspectral data is the information in the spectral details that are unavailable in lower spectral resolution data. Fused data is questionable when applied to only three spectral bands when the spatial resolution difference is small. In this case, with more spectral bands and a greater difference in spatial resolution, it is surprising that the methods work as well as they do.
It should be possible to apply the idea of HIS transformation to a 6-band data set. You've already stretched the idea from RGB to multiple channels in a narrow spectral range. It would be interesting to apply the concept to multiple spectral bands. You might have to rename it, though, to avoid confusion with color theory.
Specific Comments:
p.1: after the China-brazil → as the China-Brazil
p.2: a important → an important
Sect 2.1
p.3: at 11: 26 am → At 11:26 am
p.4: To experiment → For the experiment
p.5: the data of 452 ~ 1047 nm band → a subset of the data spanning the range 452-1047 nm was extracted from the HS data, and the bands were grouped to match specific MS bands (Table2).
Sect 2.3
p.5: inputted → input
Fig. 5: I would find it easier to relate the images to the text if there were a label (e.g., Brovey, GS, …) immediately below the corresponding image.
Fig. 6: artificial building → manmade structures (suggestion)
p.13, (1): What is color brightness? Is that similarity of the means across all bands?
Author Response
Dear Reviewer:
Thank you very much for your careful review and constructive comments with regard to our manuscript “Fusion of China ZY-1 02D Hyperspectral data and Multispectral Data: Which Methods Should Be Used?” (ID: remotesensing-1238235).
Those comments are all valuable and very helpful in revising and improving our paper, as well as guiding significance to our research. We carefully considered the comments and made corrections that we hope will be accepted. Our corrections in the manuscript and the responses to the reviewer's comments are as follows and a word file “remotesensing-1238235-revision-1” (main corrections are marked in red color) and "response-1".

Reviewer 2 Report
Authors have made an effort to answer and rectify the paper considering my previous comments. I still opine that the novelty of the contribution is low. As far as I am concerned it can be published as it except editorial considerations.
Author Response
Dear Reviewer:
Thank you very much for your careful review and constructive comments with regard to our manuscript “Fusion of China ZY-1 02D Hyperspectral data and Multispectral Data: Which Methods Should Be Used?” (ID: remotesensing-1238235).
Those comments are all valuable and very helpful in revising and improving our paper, as well as guiding significance to our research. We carefully considered the comments and made corrections that we hope will be accepted. Our corrections in the manuscript and the responses to the reviewer's comments are as follows and a word file “remotesensing-1238235-revision-1” (main corrections are marked in red color) and "response-2".

Reviewer 3 Report
It could be accepted
Author Response
Dear Reviewer:
Thank you very much for your careful review and constructive comments with regard to our manuscript “Fusion of China ZY-1 02D Hyperspectral data and Multispectral Data: Which Methods Should Be Used?” (ID: remotesensing-1238235).
Those comments are all valuable and very helpful in revising and improving our paper, as well as guiding significance to our research. We carefully considered the comments and made corrections that we hope will be accepted. Our corrections in the manuscript and the responses to the reviewer's comments are as follows and a word file “remotesensing-1238235-revision-1” (main corrections are marked in red color) and "response-3".
